# Non-Alcoholic Fatty Liver Disease: Translating Disease Mechanisms into Therapeutics Using Animal Models

**DOI:** 10.3390/ijms24129996

**Published:** 2023-06-10

**Authors:** Amina Basha, Sarah C. May, Ryan M. Anderson, Niharika Samala, Raghavendra G. Mirmira

**Affiliations:** 1Kovler Diabetes Center, Section of Adult and Pediatric Endocrinology, Diabetes and Metabolism, Department of Medicine, The University of Chicago, Chicago, IL 60637, USA; 2Department of Medicine, Division of Gastroenterology and Hepatology, Indiana University School of Medicine, Indianapolis, IN 46202, USA

**Keywords:** fatty liver, metabolism, ER stress, animal models, nutrition

## Abstract

Nonalcoholic fatty liver disease (NAFLD) is a range of pathologies arising from fat accumulation in the liver in the absence of excess alcohol use or other causes of liver disease. Its complications include cirrhosis and liver failure, hepatocellular carcinoma, and eventual death. NAFLD is the most common cause of liver disease globally and is estimated to affect nearly one-third of individuals in the United States. Despite knowledge that the incidence and prevalence of NAFLD are increasing, the pathophysiology of the disease and its progression to cirrhosis remain insufficiently understood. The molecular pathogenesis of NAFLD involves insulin resistance, inflammation, oxidative stress, and endoplasmic reticulum stress. Better insight into these molecular pathways would allow for therapies that target specific stages of NAFLD. Preclinical animal models have aided in defining these mechanisms and have served as platforms for screening and testing of potential therapeutic approaches. In this review, we will discuss the cellular and molecular mechanisms thought to contribute to NAFLD, with a focus on the role of animal models in elucidating these mechanisms and in developing therapies.

## 1. Introduction

Nonalcoholic fatty liver disease (NAFLD) is a spectrum of liver pathologies arising from excess fat accumulation in the liver in the absence of excess alcohol use or other causes of liver disease (recently reviewed in [1]). At one end of the spectrum, there is fat accumulation in the liver, or simple steatosis, which progresses to steatosis with inflammation (nonalcoholic steatohepatitis (NASH)), which can further lead to liver scarring (fibrosis). Towards the other end of the spectrum, the disease manifestation is severe, with extensive fibrosis (also known as cirrhosis) and hepatocellular carcinoma. The prevalence of NAFLD is estimated to be 30–40% worldwide and is projected to exceed 50% by 2040 [2]. This steady rise in NAFLD is likely due to an increase in its risk factors, which include obesity, type 2 diabetes mellitus, hyperlipidemia, and metabolic syndrome (MetS). Indeed, in people with type 2 diabetes mellitus and class III obesity (BMI > 40), the prevalence of NAFLD approaches 90% [3,4]. NASH and fibrosis are also more common in individuals with type 2 diabetes mellitus [5,6,7]. Based on these co-morbidities, it is thought that NAFLD is the hepatic manifestation of MetS, with insulin resistance being the main risk factor [8]. Thus, lifestyle modifications with a loss of more than 5–10% of body weight and improvement of underlying conditions can potentially reverse NAFLD before fibrosis occurs. There are currently no U.S. Food and Drug Administration (FDA)-approved drugs to treat NAFLD. Better insight into the molecular pathways that give rise to NAFLD would allow for the development of disease-targeted therapies. By studying animal models of NAFLD, researchers have uncovered the roles of insulin resistance, inflammation, oxidative stress, and endoplasmic reticulum stress in the disease progression. In this review, we discuss the roles of these pathways in NAFLD and assess the utility of various models in replicating NAFLD phenotypes, as summarized in Table 1. Additionally, we highlight treatments that have proven effective in animal models and humans and focus on potential new drug targets and important topics for future research.

## 2. Pathophysiology of NAFLD

The key drivers of NAFLD progression are insulin resistance and inflammation. In healthy individuals, free fatty acids migrate to the adipose tissue to be stored as triglycerides, and glucose is taken up by skeletal muscle for glycogenesis. However, in states of excess nutrient intake, both adipose tissue and skeletal muscle develop insulin resistance, resulting in reduced lipogenesis and increased lipolysis in adipose tissue and reduced glycogenesis in skeletal muscle, thus diverting excess substrate to the liver to be stored there as triglycerides (Figure 1). At the level of the hepatocyte, selective insulin resistance increases glucose uptake and de novo lipogenesis. Together, increased free fatty acid uptake and lipogenesis result in an overabundance of fat in the liver. In many cases, fatty liver is relatively benign and does not progress beyond this stage. However, hepatic steatosis can progress to NASH upon multiple pro-inflammatory insults from adipose tissue and the gastrointestinal tract [34]. Inflammation leads to oxidative stress and endoplasmic reticulum (ER) stress, which are key mediators of hepatocyte damage and destruction in later stages of liver disease. With this complex pathophysiology in mind, we review models of NAFLD that demonstrate obesity, insulin resistance, and inflammation in the forms of oxidative and ER stress.

## 3. Obesity and Insulin Resistance

Insulin plays an important role in glucose and lipid homeostasis [35]. In adipose tissue, insulin stimulates glucose uptake and its storage as lipid and inhibits lipolysis, thereby promoting the accumulation of lipids in adipocytes. Meanwhile, in the liver, insulin stimulates glucose storage as glycogen or synthesis into triglycerides that are packaged and exported into circulation. Obesity leads to insulin resistance, thereby disrupting these critical pathways that maintain lipid homeostasis [36]. This is, in part, due to adipose tissue release of pro-inflammatory cytokines that inhibit insulin signaling [37]. Upon insulin resistance, adipocytes release free fatty acids that accumulate in the liver, leading to hepatic steatosis [38]. Thus, obesity and insulin resistance are important hallmarks of NAFLD. Purely genetic causes of NAFLD are rare, but deficiencies in these key genes can serve as suitable animal models of NAFLD. Otherwise, dietary interventions are used to generate animal models of NAFLD.

### 3.1. Genetic Models of Obesity and Insulin Resistance

Leptin is a satiety hormone released by adipose tissue to suppress appetite [39]. It functions to reduce caloric intake, increase hepatic triglyceride export, and block de novo lipogenesis [40,41]. Therefore, leptin-deficient (*ob*/*ob*) mice are an important model of NAFLD. *Ob*/*ob* mice become obese and develop high fasting blood glucose with insulin resistance [9,10]. Importantly, *ob*/*ob* mice show mild to severe steatosis but do not progress to NASH without additional stimuli, such as a diet containing high cholesterol, trans fat, and fructose (known as the Amylin mouse liver NASH model, or AMLN, diet) [19,42]. Leptin receptor-deficient (*db*/*db*) mice are similar to *ob*/*ob* mice in terms of obesity, but they show less steatosis than *ob*/*ob* mice [11]. Similarly, Zucker (*fa*/*fa*) rats have a spontaneous mutation in the leptin receptor that leads to hyperphagia, severe obesity, and insulin resistance [12]. Because human mutations in leptin/leptin receptor are rare [43,44] and because leptin levels increase with NAFLD severity [45], these models may not accurately represent the pathogenesis of NAFLD.

Another gene involved in satiety is *Alms1* [46]. Mutations in *Alms1* are responsible for Alström syndrome, a rare genetic disorder leading to childhood obesity, severe insulin resistance, and multiple organ failure [47]. Many of these individuals develop an accelerated form of NAFLD, leading to fibrosis and cirrhosis at an unexpectedly young age [48]. In mice, a truncation mutant of Alms1 (*foz*/*foz*) leads to increased weight gain, insulin resistance, type 2 diabetes mellitus, and steatosis [13]. On a high-fat diet, *foz*/*foz* mice develop NASH due to hepatic cholesterol accumulation [49]. Mutations in *Alms1*, like mutations in leptin, are rare in humans. Therefore, there may be key differences in the pathogeneses of NAFLD in humans and *foz*/*foz* mice.

Insulin signaling activates sterol regulatory-element-binding proteins (SREBPs), which are master transcriptional regulators essential for maintaining lipid homeostasis pathways [50]. During insulin resistance, increased plasma insulin levels may lead to overstimulation of these pathways. SREBP-1c is known to upregulate fatty acid synthesis genes [51]. When SREBP-1c is overexpressed in adipose tissue, mice develop hyperglycemia, insulin resistance, and fatty liver [4]. Insulin also activates SREBP-2, thereby activating genes involved in cholesterol synthesis [52]. SREBP-2 is shown to be upregulated during NASH [53]. Therefore, SREBP-2 may provide a direct link between insulin resistance, cholesterol accumulation, and NAFLD.

It is now recognized that excess hepatic cholesterol, in addition to excess triglycerides, promotes NAFLD progression [54]. Excess hepatic cholesterol may result from increased biosynthesis, impaired very-low-density lipoprotein (VLDL) packaging and export, or impaired biliary excretion. Thus, animal models with defects in cholesterol metabolism are now being used to study NAFLD pathogenesis. Mice deficient in low-density lipoprotein receptor (*Ldlr*^−/−^), when fed a high-fat/high-cholesterol diet for 3 months, develop hepatic inflammation, steatosis, and fibrosis [14]. Apolipoprotein E-deficient (*Apoe*^−/−^) mice fed a high-fat diet progress to early stages of fibrosis after 8 weeks [55]. In these models, excess free cholesterol accumulation in hepatocytes is likely the main driver of hepatic inflammation.

### 3.2. Dietary Models of Obesity and Insulin Resistance

Dietary deficiency in choline or defects in methionine metabolism may lead to NAFLD. Choline, an essential nutrient, is required for the synthesis of phosphatidylcholine, which is the main phospholipid on VLDL particles [56]. In the absence of phosphatidylcholine, excess hepatic triglycerides accumulate, as they cannot be packaged into VLDL and exported into circulation. Therefore, in mice, choline-deficient diets augment hepatic lipid accumulation on a high-fat diet [20]. However, their liver disease does not go beyond this stage unless they are fed a diet that is also deficient in methionine. Methionine, an essential amino acid, is another precursor of phosphatidylcholine, acting via methylation of phosphatidylethanolamine. Mice fed a methionine-choline-deficient (MCD) diet show weight gain, insulin resistance, and liver inflammation within 4 weeks [57]. However, this model does not fully mimic the human pathophysiology, as the mice do not develop insulin resistance [58]. Genetic models that interfere with the utilization of methionine—such as an S-adenosylhomocysteine hydrolase mutant in zebrafish (*ducttrip*) [17,59] or methionine adenosyltransferase 1A (MAT1A) deficiency in mice [18]—have also been useful in studying NAFLD.

A high-fat, high-carbohydrate, and/or high-cholesterol diet effectively induces insulin resistance in animal models, triggering NAFLD without genetic manipulation [60]. Many diet variations exist, but the typical range is 32–60% fat, 30–50% sugar, and 0.2–1% cholesterol [60]. In Sprague–Dawley rats, a high-fat diet (60% fat, 20% carbohydrates) or NASH diet (40% fat, 40% carbohydrates, 2% cholesterol) induces steatosis and inflammation after 16 weeks [22]. On a high-fructose diet (10% fat, 70% carbohydrates), the rats do not show the same phenotypes, suggesting that fat and cholesterol are the main drivers of NAFLD progression [22]. On the American lifestyle-induced obesity syndrome (ALIOS) diet, which consists of 45% fat (of which 30% is trans fat) plus high-fructose corn syrup in drinking water, mice develop obesity, insulin resistance, steatosis, NASH, fibrosis, and hepatic tumors over a 12-month period [23]. Thus, the ALIOS model replicates the human pathophysiology with striking accuracy. Similarly, in the diet-induced animal model of non-alcoholic fatty liver disease (DIAMOND), mice fed a high-fat, high-carbohydrate diet (42% fat, 0.1% cholesterol, high-fructose/glucose water) develop steatosis, NASH, advanced fibrosis, and liver tumors after 52 weeks [24]. When fed a fast-food diet of 40% fat (12% saturated fats) and 0.2% cholesterol plus high-fructose corn syrup in drinking water for 25 weeks, mice show effects that are physiologically comparable to humans, including NASH with fibrosis and markers of cellular stress [25]. However, in this case, liver tumors were not reported [25]. In many of these studies, the length of time to advanced disease is a major limitation. To accelerate disease progression, SMC Laboratories developed a mouse model (STAM™) that progresses to liver tumors in 100% of mice by 20 weeks [26]. STAM™ mice are first injected with the β-cell toxin streptozotocin to induce type 2 diabetes. Subsequently, the mice are fed a high-fat diet to induce liver disease. In conclusion, dietary models show the most phenotypic similarity to human NAFLD, although a longer feeding period may be required to reproduce more severe or advanced stages of liver disease, such as cirrhosis and liver tumors.

## 4. Inflammation

Obesity and insulin resistance cause hepatic steatosis, but additional pathogenic insults are required to progress to NASH. Inflammation drives the progression of NAFLD from benign fat accumulation to permanent liver damage. Two major sources of inflammation in NAFLD are oxidative stress and ER stress.

### 4.1. Oxidative Stress

Reactive oxygen species (ROS) are oxygen-containing molecules that can readily react with and damage biomolecules. Whereas low levels of ROS act as key signaling messengers, excess ROS promote oxidative stress. In NAFLD, the accumulation of lipids results in excess ROS, inflammation, and, ultimately, progression to NASH [61,62]. Both increased ROS production and impaired antioxidant capacity contribute to elevated ROS.

Most ROS are produced during mitochondrial oxidative phosphorylation, as electrons escape the electron transport chain, react with oxygen, and form superoxide [63]. During NAFLD, hepatocytes upregulate β-oxidation to break down excess fatty acids [64,65]. With this surge in β-oxidation, more reducing equivalents are shuttled into the electron transport chain, thereby increasing superoxide production and resulting in oxidative stress and inflammation [66,67]. Impaired activity of the electron transport chain complexes further exacerbates ROS production and oxidative stress. In NASH, the activity of all electron transport chain complexes is impaired [68]. Another major source of ROS is the mitochondrial cytochrome P450 (CYP) enzyme CYP2E1, which is elevated in NASH livers [69]. CYP2E1 metabolizes alcohol and, in the process, generates ROS [70]. In mice, CYP2E1 promotes the development of obesity, insulin resistance, NASH, and fibrosis on a high-fat diet [71,72,73]. In some studies, free fatty acids were shown to upregulate CYP2E1 [74,75,76], but other studies contradict these findings [77,78]. NADPH oxidases (NOXs) are another source of ROS. NOXs make superoxide at relatively low levels to support normal physiological functions, such as inflammatory responses and cell signaling [79]. During NAFLD, NOXs may become dysregulated and produce excessive amounts of ROS. In the liver, NOX1, NOX2, and NOX4 are thought to be the most relevant isoforms that contribute to disease pathogenesis. Indeed, NOX1- or NOX4-deficient mice are protected from liver fibrosis induced by carbon tetrachloride [80]. Similarly, NOX2-deficient mice are protected against high-fat diet-induced liver steatosis [81].

In NAFLD, excess ROS can also result from the failure of antioxidant systems. When high amounts of ROS accumulate, they deplete antioxidants and overwhelm the liver’s antioxidant enzymes. A key antioxidant pathway involves the redox-sensitive transcription factor nuclear factor erythroid 2-related factor 2 (Nrf2), which binds to antioxidant response elements and upregulates detoxifying enzymes and the antioxidant glutathione [82]. Nrf2 plays an important protective role in NAFLD, as Nrf2-deficient mice fed a high-fat diet are more susceptible to NASH [83]. Conversely, activation of the Nrf2 pathway helps to resolve NASH and fibrosis in mice fed an MCD diet [84]. Nrf2 upregulates detoxifying enzymes, including NAD(P)H:quinone oxidoreductase 1 (NQO1), heme oxygenase-1 (HO-1), glutathione S-transferase (GST), and superoxide dismutase (SOD), and these have been shown to protect against NAFLD progression to varying degrees. In mice, overexpression of NQO1 protects against high-fat diet-induced steatosis [85]. Inducing HO-1 activity protects against MCD diet-induced steatohepatitis [86]. Loss of GST (mu 2 isoform) results in hepatosteatosis and fibrosis in mice fed a high-fat diet [87]. Loss of SOD1 increases hepatic lipids and oxidative damage and eventually leads to hepatocellular carcinoma [88,89]. Nrf2 also upregulates peroxisome proliferator-activated receptor γ (PPARγ) [90,91,92], a transcription factor with anti-inflammatory, antioxidant, and insulin-sensitizing functions [93]. In adipose tissue, PPARγ promotes lipid uptake and storage, as well as insulin sensitivity [92]. Additionally, in hepatic macrophages, PPARγ protects mice against oxidative stress induced by the liver toxin carbon tetrachloride [28]. These potential benefits have led to increased interest in using PPARγ agonists to treat NAFLD. However, because liver PPARγ increases free fatty acid uptake and de novo lipogenesis, its expression in the liver promotes steatosis [94,95,96]. Therefore, cell-specific targeting of PPARγ would be essential. The presence of non-enzymatic antioxidants, including vitamins C and E, has been inversely associated with NAFLD/NASH, suggesting that they also play a role in preventing liver disease [97,98].

#### Animal Models of Oxidative Stress in NAFLD

To study the role of oxidative stress in NAFLD, mice are commonly fed an MCD diet. This diet appears to induce more inflammation, ROS, mitochondrial DNA damage, and apoptotic cell death than in *ob*/*ob* mice [99]. However, there are key differences between the MCD diet model and the human pathophysiology of NAFLD. Most notably, mice on an MCD diet become cachectic with a reduced liver-to-body weight ratio and have elevated liver enzymes, which are not observed in humans with NASH [21]. These key differences lead us to question whether the MCD diet model is suitable for studying NAFLD. Zucker rats (*fa*/*fa* leptin-deficient model) also show increased markers of oxidative stress when fed a high-fat diet [100]. However, these *fa*/*fa* rats show decreased CYP2E1 with no change on a high-fat diet [100], which is the opposite response of that seen in humans [69]. In zebrafish, overexpression of proinflammatory cytokines (IL-1β, TNFα, and IFNγ) increases inflammation, promotes macrophage infiltration into hepatocytes, and generates excess ROS [29]. Additionally, this model causes lipid accumulation in hepatocytes and signs of both insulin resistance and inflammation, similar to the human pathophysiology of NAFLD [29].

### 4.2. ER Stress

Hepatocytes are highly secretory cells that produce a variety of proteins to maintain cellular, metabolic, and lipid homeostasis. These proteins must be properly synthesized, processed, and folded in the hepatocyte ER. Anything that disrupts ER function can induce ER stress and the accumulation of unfolded or misfolded proteins. In NAFLD, many factors drive ER stress in hepatocytes, including free cholesterol accumulation in the ER membrane [101]. In response, hepatocytes first attempt to reduce ER stress and promote cell survival, but if unsuccessful they undergo apoptosis. This process is called the unfolded protein response (UPR), a complex signaling pathway with three stress-activated sensors: protein kinase R-like ER kinase (PERK), activating transcription factor 6 (ATF6), and inositol-requiring enzyme 1α (IRE-1α). Chronic activation of the UPR is thought to contribute to NAFLD.

The master regulator of the UPR is a chaperone protein, glucose-regulated protein 78 (GRP78). Underscoring its protective role in NAFLD, GRP78 overexpression in the liver of *ob/ob* mice protects against steatosis [102]. Under unstressed conditions, GRP78 binds to the three UPR sensors, maintaining them in an inactive state. During ER stress, GRP78 preferentially binds to the unfolded proteins. Released from GRP78, the UPR sensors enter an active state. Once activated, PERK phosphorylates eukaryotic initiation factor 2α (p-eIF2α), which shuts down global protein translation to ameliorate ER stress but also upregulates activating transcription factor 4 (ATF4). Initially, ATF4 activates genes that promote cell survival, but if ER stress is severe or prolonged ATF4 upregulates pro-apoptotic C/EBP homologous protein (CHOP). Consistent with a pathogenic role in NAFLD, ATF4-deficient mice are protected against high-carbohydrate diet-induced steatosis [103], and ATF4-overexpressing zebrafish developed steatosis [104]. Activated ATF6 is proteolytically cleaved into its nuclear form, which then upregulates genes involved in protein folding and lipid synthesis. It was shown that ATF6 deficiency protects against steatosis in a zebrafish model of chronic hepatic ER stress (i.e., foie gras mutant), whereas it may potentiate steatosis under acute ER stress [32]. IRE1α has endonuclease activity, which it uses to splice *Xbp1* mRNA, producing an active transcription factor. Spliced XBP1 activates the transcription of ER chaperones [105] and ER-associated protein degradation components [106] to promote cell survival. Multiple reports have shown that IRE1α protects against hepatic steatosis [107] via UPR-unrelated pathways that regulate lipid homeostasis [108,109]. Under severe stress, IRE1α couples with TNF receptor-associated factor 2 (TRAF2) to activate c-Jun N-terminal kinase (JNK) [110] or IκB kinase (IKK) [111], thereby promoting inflammation. More recently, IRE1α was shown to be involved in NLRP3 inflammasome assembly and production of the pro-inflammatory cytokine IL-1β [112]. Therefore, under the right conditions, IRE1α signaling may promote progression from simple steatosis to NASH.

When the UPR fails to relieve ER stress, it triggers apoptosis through various proteolytic caspases, potentially including a combination of caspases 12, 3, 6, 7, 8, and 9 [113]. Another caspase (caspase 2) may be activated by ER stress, but is not believed to play a role in ER stress-mediated apoptosis [114]. Instead, in the *MUP-uPA* mouse model of hepatic ER stress, caspase 2 cleaves site-1 protease, thereby activating SREBPs [115]. As discussed, the SREBPs are master regulators of lipid homeostasis and play a role in NAFLD progression. In a fructose-induced mouse model of NAFLD, studies have linked ER stress to SREBP-1c activation and lipid accumulation [116]. Inhibiting SREBP-2 overactivation in vitro improves hepatic autophagy and relieves ER stress in mice fed a high-fat diet [117]. However, activating SREBP-2 in mice with lovastatin/ezetimibe improves autophagy and reduces hepatic triglyceride accumulation [118]. More work is needed to assess the role of SREBP-2 in NAFLD and progression to NASH. Moreover, additional studies must disentangle when the UPR is protective versus when it promotes NAFLD progression, which is an essential first step in developing NAFLD therapeutics to target ER stress.

#### Animal Models of ER Stress in NAFLD

To that end, we examined select animal models that may prove useful in studying ER stress as a pathogenic step towards NAFLD. As previously mentioned, the major urinary protein urokinase plasminogen activator (*MUP-uPA*) transgenic mouse is a model of ER stress and NAFLD progression [119]. The protein uPA is delivered to hepatocytes, where it accumulates in the ER and induces ER stress [30]. After 24 weeks on a high-fat diet, the mice develop NASH, and after 32 weeks, they progress to hepatocellular carcinoma [30]. The *MUP-uPA* mouse model is one of the better models of NAFLD progression owing to its ability to faithfully replicate key human phenotypes throughout the entire disease progression [120]. In zebrafish, deletion of the trafficking protein particle complex subunit 11 (TRAPPC11) analogue (also known as the foie gras gene) disrupts ER and Golgi protein trafficking and induces ER stress [31]. The foie gras model results in hepatic steatosis mediated through the ATF6 branch of the UPR [32]. However, because this mutation can be lethal, the foie gras model progresses neither to liver fibrosis nor to hepatocellular carcinoma. Therefore, this model does not demonstrate full disease pathogenesis. Also in zebrafish, deficiency in CDP-diacylglycerol-inositol 3-phosphatidyltransferase (Cdipt) impairs phosphatidylinositol synthesis, leading to ER stress and hepatic steatosis by an unknown mechanism [33].

## 5. Treatment Modalities

The mainstay treatment for NAFLD is lifestyle modification. Beyond diet and exercise, there are no current FDA-approved or guideline-approved medications to treat NAFLD. Potential new therapeutics target the underlying causes of NAFLD, which include insulin resistance, imbalances in lipid metabolism, and inflammation.

### 5.1. Lifestyle Modifications

The American Association for the Study of Liver Diseases (AASLD) recommends a body weight reduction of at least 3–5% to reverse steatosis and >7% to improve histologic features of NASH and fibrosis [121]. The European Association for the Study of the Liver (EASL) guidelines are similar and recommend a weight reduction of 7–10% for treatment of NAFLD [122]. For a healthy eating plan that manages MetS, the American Heart Association (AHA) recommends reducing saturated fat intake to <7% of total calories, minimizing trans fat intake, and maintaining cholesterol intake at <200 mg per day and total fat at 25–35% of total calories [123]. For an exercise plan, individuals should aim for at least 30 min of cardio activity at least 5 days a week [123].

Recent randomized controlled clinical trials have investigated the impacts of diet composition and meal timing on NAFLD. In the REDUCTION trial, subjects with type 2 diabetes mellitus and NAFLD lost weight and improved their glycemic control after 6 months on a low-carbohydrate, high-fat diet compared to patients on a high-carbohydrate, low-fat diet [124]. However, these improvements were not sustained at the 3-month follow-up [124]. Long-term adherence often limits the effectiveness of dietary interventions. Although low-fat diets and Mediterranean diets (low carbohydrate and rich in unsaturated fats) have both been shown to reduce hepatic steatosis in clinical trials, adherence was better for the Mediterranean diet [125], suggesting the longer-term benefits of a Mediterranean diet. Moreover, adherence to a Mediterranean diet has been shown to lower liver inflammatory markers and increase antioxidant pathways [126]. In a US-based study, the effects of alternate-day fasting with and without exercise were tested on subjects with obesity and NAFLD [127]. For 3 months, study participants ate one meal at dinnertime on “fast days” and ate food as desired on “feast days”. At the end of the trial, participants in the group of alternative-day fasting plus aerobic exercise had decreased body weight, reduced steatosis, and improved glycemic control [127]. However, more studies are needed to determine whether exercise provided any benefit over fasting alone. Lastly, the TREATY-FLD trial recently found no additional benefit with time-restricted diets (eating between 8:00 a.m. and 4:00 p.m.) as opposed to a calorie-restricted diet alone [128]. After 12 months, participants in both groups had reduced and comparable intrahepatic fat measurements [128].

Research has clearly shown that exercise improves NAFLD, even without weight loss [129]. Despite NAFLD patients preferring exercise over medication, many are unable to start or maintain exercise programs due to barriers such as fatigue, injury, and shortness of breath [130]. As NAFLD progresses, these barriers are even more difficult to overcome. Thus, there is a need to tailor exercise programs (in terms of exercise type, duration, and intensity) to address these limitations. A comparative analysis of aerobic and resistance exercise found that both forms of exercise improved hepatic steatosis [131]. However, because resistance exercise requires less energy consumption, it may be the better choice for patients who cannot tolerate aerobic exercise [131].

### 5.2. Drugs Targeting Insulin Resistance

Thiazolidinediones (TZDs) are a class of drugs that improve insulin sensitivity by activating PPARγ [132]. Two TZDs, pioglitazone and rosiglitazone, are FDA approved to treat insulin resistance and improve glycemic control in patients with type 2 diabetes mellitus. Unfortunately, these drugs may lead to heart failure [133,134] and weight gain [135]. Thus, there has been interest in developing substitutes with better safety profiles. Recently, dual PPARα/γ agonists have shown promise in treating NAFLD/NASH, as they not only improve glycemic control (via PPARγ) but also reduce lipid levels (via PPARα). Thus, they could target multiple underlying factors of NAFLD pathogenesis while also preventing side effects such as weight gain. In preclinical studies of *Apoe^−/−^* mice on a high-fat/high-cholesterol diet, the dual PPARα/γ agonist aleglitazar improved glucose tolerance and lowered hepatic fat content without an increase in body weight [136]. In the AleCardio trial, aleglitazar reduced hepatic steatosis and fibrosis in subjects with acute coronary syndrome and type 2 diabetes mellitus [137]. Saroglitazar, another dual PPARα/γ agonist, showed similar benefits in mouse models of NAFLD/NASH [138] and in a recent phase 2 clinical trial [139]. However, in the clinical trials, aleglitazar and saroglitazar increased body mass index and body weight, respectively. A third PPAR family member, PPARδ, was recently shown to reduce hepatic lipid content via autophagy [140]. Therefore, it was thought that pan-PPAR agonists (targeting PPARα, PPARγ, and PPARδ) may show additional benefits over more selective agonists. In preclinical studies of mice fed a choline-deficient, amino acid-defined, high-fat diet, the pan-PPAR agonist lanifibranor was more potent than single agonists in improving steatohepatitis and carbon tetrachloride-induced fibrosis [141]. In the NATIVE trial, the pan-PPAR agonist lanifibranor improved advanced fibrosis in subjects with type 2 diabetes mellitus [142]. However, much like the more selective PPAR agonists, lanifibranor also induced weight gain [142]. More studies will be needed to assess the possible benefits of pan-PPAR agonists over dual agonists and weigh the risks of potential side effects.

Sodium-glucose cotransporter 2 (SGLT2) inhibitors prevent glucose from being reabsorbed into circulation by the kidneys, thus improving hyperglycemia [143]. SGLT2 inhibitors result in glucose excretion in urine, and the loss of these calories may be one reason why SGLT2 inhibitors induce weight loss [144]. Various SGLT2 inhibitors have been FDA approved to treat patients with type 2 diabetes mellitus, but they are just now being investigated in the context of NAFLD. In a small clinical trial, the FDA-approved SGLT2 inhibitor dapagliflozin improved glycemic control and hepatic lipid content after 12 weeks of treatment [145]. Similarly, ipragliflozin (a non-FDA-approved drug) helped to resolve NASH and fibrosis in subjects with type 2 diabetes and NAFLD after 72 weeks [146]. Preclinical studies tested the combined effects of tofogliflozin (an SGLT2 inhibitor) and pioglitazone (a PPARγ agonist) on animal models of obesity and type 2 diabetes [147]. Interestingly, the researchers found that tofogliflozin prevented pioglitazone-induced weight gain in mice, and the combination therapy improved hyperglycemia better than monotherapy [147]. These effects were also observed in a clinical trial investigating tofogliflozin/pioglitazone combination therapy, as was a further improvement in steatosis over tofogliflozin monotherapy in patients with type 2 diabetes and NAFLD [148]. These studies used small cohorts, ranging from 20–40 participants, so larger-scale studies will need to be conducted to understand the effects of these medications on NAFLD.

Glucagon-like peptide-1 (GLP-1) analogs are another class of drug commonly used to treat type 2 diabetes. These drugs, like GLP-1, stimulate pancreatic β cells to secrete insulin. The GLP-1 receptor agonist liraglutide has shown some promise in decreasing hepatic fat content in subjects with obesity and/or type 2 diabetes [149,150]. However, a recent meta-analysis concluded that the effects of liraglutide were nonsignificant in individuals with NAFLD [151]. Another GLP-1 receptor agonist, semaglutide, has a similar mechanism of action to that of liraglutide but with additional weight loss and reductions in glycated hemoglobin (HbA1c) [152]. In a phase 2 clinical trial, semaglutide resolved NASH in up to 59% of study participants, but its effects on fibrosis outcomes were less clear [153]. There is ongoing research to understand the mechanisms by which GLP-1 analogs may improve steatosis and fibrosis.

### 5.3. Drugs Targeting Lipid Metabolism

Cholesterol is now recognized for its role in NAFLD pathogenesis. Many individuals with NAFLD also have dyslipidemia and are prescribed statins to lower their cholesterol levels. Additionally, statins may have anti-inflammatory functions beyond their primary role in lowering cholesterol [154]. Thus, statins could be effective treatments for NAFLD. In the ESSENTIAL trial, a combination of the lipid-lowering agents ezetimibe (a non-statin) and low-dose rosuvastatin was shown to significantly reduce hepatic fat content in participants with NAFLD [155]. However, longer-term studies will be needed to determine whether lipid-lowering therapies can improve NASH and/or fibrosis.

Farnesoid X receptor (FXR) is a nuclear transcription factor that increases bile acid elimination and reduces hepatic triglyceride levels, among various other metabolism-benefiting functions [156]. Therefore, FXR agonists show promise as therapeutics for NAFLD. In recent clinical trials, the FXR agonists MET409 [157], tropifexor [158], vonafexor [159], and cilofexor [160] were shown to reduce hepatic fat content in subjects with NASH. In an ongoing phase 3 clinical trial, the FXR agonist obeticholic acid improved fibrosis in people with NASH and moderate to severe fibrosis [161]. Currently, a new drug application for obeticholic acid is under review by the FDA for treatment of NASH with advanced fibrosis. Common side effects of FXR agonists include itching and/or increased low-density lipoprotein cholesterol levels.

Fibroblast growth factor 21 (FGF21) is another broad metabolic regulator that is being considered as a treatment for NAFLD. In mice fed a high-fat diet, recombinant murine FGF21 upregulates genes involved in fatty acid oxidation and downregulates genes involved in lipogenesis, thus leading to a reduction in hepatic steatosis [162]. Multiple clinical trials have shown that FGF21 analogs, including efruxifermin [163], LLF580 [164], and pegozafermin [165], reduce hepatic fat content and lead to weight loss. Because these FGF21 analogs have such a broad range of functions, long-term safety and efficacy studies will be critical.

More recent studies are investigating thyroid hormone receptor-β (TR-β) agonists as potential NASH treatments. Synthetic thyroid hormone, or thyroxine, has been shown to increase lipid metabolism, leading to weight loss [166]. On the other hand, thyroxine can also lead to heart failure and arrythmias [167]. By targeting the predominant liver isoform (TR-β), hepatic lipid content can be reduced without adverse cardiac effects [168]. Several TR-β agonists have shown efficacy for NAFLD treatment in animal models [169]. In a phase 2 clinical trial, the TR-β agonist resmetirom was shown to reduce hepatic fat in individuals with NASH [170]. Although promising, larger studies are still needed to demonstrate the efficacy and safety of TR-β agonists.

### 5.4. Drugs Targeting Inflammation

Pentoxyfylline is an FDA-approved anti-inflammatory drug that blocks the production of pro-inflammatory cytokines such as tumor necrosis factor α (TNFα) [171]. In MCD diet-induced NASH mice, pentoxyfylline decreases TNFα mRNA expression and markers of hepatic inflammation [172]. However, pentoxyfylline-treated mice have increased hepatic steatosis [172]. Despite this observation, a meta-analysis concluded that pentoxyfylline decreases steatosis, as well as fibrosis and liver inflammation [173]. Larger studies are needed to confirm these effects.

Vitamin E is a key antioxidant with anti-inflammatory functions [174]. Based on its ability to suppress oxidative stress, vitamin E has also been considered as therapy for NAFLD. Indeed, in mice on an MCD diet, vitamin E reduces hepatic steatosis and markers of hepatic inflammation [175]. Additionally, the PIVENS trial showed improvement in steatosis and inflammation in non-diabetic NASH participants treated with vitamin E compared to placebo [176]. There have been some concerns about vitamin E safety in the long term, owing to possible links to hemorrhagic stroke and prostate cancer [121]. Nonetheless, the AASLD guidelines advise that individuals with severe fibrosis and without type 2 diabetes may consider taking a vitamin E supplement [121].

### 5.5. The Future of Preclinical Testing: Nonhuman Primates as Models of NAFLD

Many drug candidates never succeed in clinical trials because the drug’s effects in mice and other model systems do not translate to humans. Thus, there is a need for model systems that better reflect the human pathogenesis of NAFLD. Nonhuman primates have the potential to more accurately predict how a drug will behave in human studies. NAFLD has been shown to occur in various nonhuman primates spontaneously [177] or as a natural result of aging [178]. However, in these animals, the disease can be accelerated or worsened by feeding special diets. Recently, a new dietary model of NASH in cynomolgus monkeys was developed by feeding a high-fat, high-fructose, high-cholesterol diet for 10 months [27]. In monkeys, this diet induces hepatic steatosis, inflammation, NASH, and fibrosis that is strikingly similar to the human NAFLD pathogenesis at the transcriptional level [27]. In this model system, the tripeptide DT-109 (Gly-Gly-Leu) attenuates NASH and fibrosis by upregulating fatty acid degradation and downregulating inflammation [27]. An important caveat in these studies is their use of only male animals, thereby limiting applicability across the sexes.

## 6. Conclusions: Translation from Animals to Humans

The demand for animal models to study the pathogenesis of NAFLD is more vital now, as the prevalence of NAFLD is increasing worldwide and therapeutic candidates need to be identified and tested. Given the plethora of research that has been done on animal models to understand the pathophysiology of NAFLD, we now have a better understanding of the various insults that occur in the progression of disease. This has allowed for the implementation of specific guidelines for lifestyle modifications and potential pharmacotherapy. Of all the animal models described here (Table 1), the high-fat/high-fructose diet model has proven to be most like the human pathophysiology of disease. The hypothesized mechanisms describing the causal role of fructose in the development of NAFLD include direct up-regulation of de novo lipogenesis enzymes by fructose breakdown products in hepatocytes and intestinal wall weakening, leading to increased uptake of endotoxins from the gut to the liver and driving inflammation. However, most preclinical studies have used male rodent models, making it is difficult to determine whether sexual dimorphism might preclude applicability to females. Furthermore, these dietary changes alone are usually insufficient for demonstrating severe disease. Although some models, such as *fa/fa* rats or MCD diet-fed mice, show similar pathophysiology to humans, many of them lack a key clinical feature of obesity that is seen in humans. The *MUP-uPA* transgenic mouse fed a high-fat diet may be the best animal model of ER stress leading to NAFLD. However, many of the studies discussed earlier were only completed once, leading to the question of reproducibility. Therefore, animal models to date still have limitations that do not fully recapitulate human disease.

Taken together, whereas animal models have provided crucial pathogenic insight into the multiple insults contributing to NAFLD, they only provide a first step towards identifying new therapies. The further development of non-human primate models and/or more rigorous studies in humans will be needed to move the field forward. The most representative animal models must also feature multiple pathogenic insults in order to fully mimic human NAFLD and reveal new therapeutic strategies. In addition to the pathogenic insults described in this review, viral hepatitis may also trigger steatosis and/or accelerate NAFLD progression, but the mechanisms are just beginning to be explored. In the future, a multiple-step treatment program may be necessary to fully treat both the factors that lead to MetS and the inflammatory and oxidative stress pathways that lead to NAFLD.

## Figures and Tables

**Figure 1 ijms-24-09996-f001:**
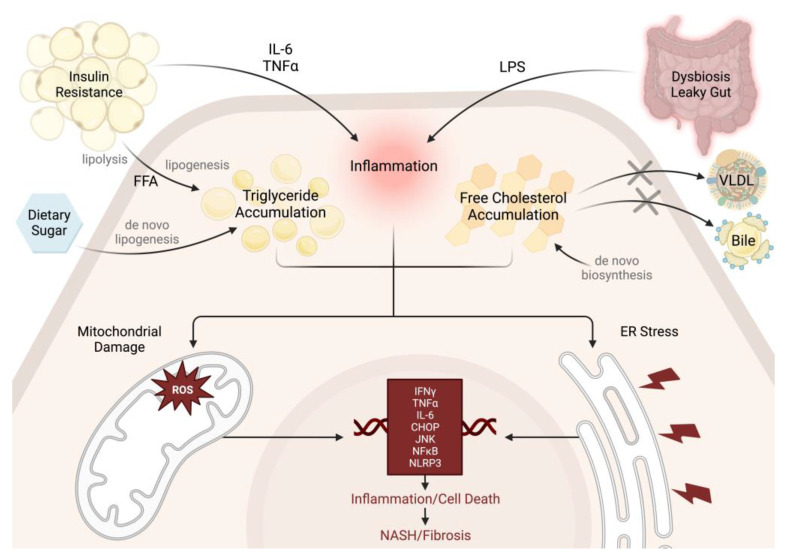
Pathogenesis of NAFLD. Insulin resistance stimulates the breakdown of adipose triglycerides into free fatty acids (FFA) that enter the liver and accumulate there as triglycerides. Excess dietary sugars are converted into additional triglycerides via de novo lipogenesis. Pro-inflammatory cytokines from the adipose tissue and lipopolysaccharides (LPS) from the gastrointestinal tract contribute directly or indirectly to hepatic inflammation. Cholesterol accumulates in the liver due to impaired very-low-density lipoprotein (VLDL) secretion and bile formation, as well as increased de novo biosynthesis of cholesterol. These cellular insults trigger the mitochondria to produce toxic levels of reactive oxygen species (ROS). Additionally, cellular insults activate endoplasmic reticulum (ER) stress and the unfolded protein response. Together, these pathways upregulate pro-inflammatory cytokine expression (IFNγ, TNFα, and IL-6), apoptotic mediators (CHOP and JNK), and immune-response mediators (NFκB and NLRP3). Ultimately, this results in hepatic inflammation and cell death, leading to NASH and fibrosis. Image created with Biorender.com.

**Table 1 ijms-24-09996-t001:** Animal models of nonalcoholic fatty liver disease (NAFLD).

Background	Model	Description	Phenotype	Limitations
**Genetic models of obesity and insulin resistance**				
Mouse	*ob/ob* [9,10]	Spontaneous mutation in leptin gene (*ob*)	Obesity Insulin resistance Steatosis	Leptin mutations rare in humans, require second stimulus (high-fat diet) to progress beyond steatosis
Mouse	*db/db* [11]	Spontaneous mutation in leptin receptor gene (*db*)	Obesity Insulin resistance Steatosis	Leptin receptor mutations rare in humans, require second stimulus (high-fat diet) to progress beyond steatosis
Rat (Zucker)	*fa/fa * [12]	Spontaneous mutation in leptin receptor gene (*fa*)	Obesity Insulin resistance Steatosis	Leptin mutations rare in humans, require second stimulus (high-fat diet) to progress beyond steatosis
Mouse (NOD)	*foz/foz * [13]	Mutation in *Alms1* gene	Obesity Insulin resistance High cholesterol Steatosis	Requires second stimulus (high-fat diet) to develop NASH, fibrosis
Mouse (B6SJLF1/J)	aP2-nSREBP-1c transgenic [4]	Overexpression of SREBP-1c in adipose tissue	Insulin resistance Steatosis Lipodystrophy	No evidence of fibrosis, liver tumors
Mouse (C57BL/6J)	*Ldlr*^−/−^ [14,15]	Targeted mutation in *Ldlr* gene encoding low-density lipoprotein receptor + high-fat/high-cholesterol diet (3 months)	Steatosis Liver inflammation Fibrosis High cholesterol	Mutation does not induce insulin resistance
Mouse (C57BL/6J)	*Apoe*^−/−^ [16]	Targeted mutation in *Apoe* gene encoding apolipoprotein E + high-fat/high-cholesterol diet (7 weeks)	Mild obesity Insulin resistance Steatosis NASH Fibrosis	Less pronounced obesity
Zebrafish	Ducttrip (dtp) [17]	Mutation in *ahcy* gene encoding S-adenosylhomocysteine hydrolase	Increased pro-inflammatory cytokines Steatosis	Recessive lethal, does not induce NASH, fibrosis, or liver tumors
Mouse (C57BL/6J)	*Mat1a*^−/−^ [18]	Mutation in *Mat1a* gene encoding methionine adenosyltransferase 1A	Steatosis Liver inflammation	No evidence of metabolic syndrome (MetS) or fibrosis
**Dietary models of obesity and insulin resistance**				
Mouse (C57BL/6 or *ob*/*ob)*	Amylin liver NASH (AMLN) [19]	High-fat/high-fructose diet (40% fat, 22% fructose, 2% cholesterol; 8 weeks)	Obesity Steatosis NASH Fibrosis	No evidence of liver tumors
Mouse (C57BL/6)	Choline-deficient + high-fat diet [20]	Choline-deficient/high-fat diet (45% fat; 8 weeks)	Obesity Steatosis	Choline-deficient diet improves high-fat diet-induced insulin sensitivity, no evidence of NASH or more advanced stages
Mouse (C57BL/6)	Methionine–choline-deficient (MCD) diet [21]	Standard chow diet deficient in methionine and choline (15 days)	Weight loss Steatosis NASH Fibrosis	Some phenotypes are the opposite of humans with NAFLD (i.e., model induces weight loss, insulin sensitivity)
Rat (Sprague–Dawley)	NASH diet [22]	High-fat/high-fructose/high-cholesterol diet (40% fat, 40% carbohydrate, 2% cholesterol; 16 weeks)	Obesity Steatosis Fibrosis	No evidence of liver tumors
Mouse (C57BL/6N)	American lifestyle-induced obesity syndrome (ALIOS) [23]	High-fat/high-carbohydrate diet (45% fat, 55% fructose/45% glucose in drinking water; 26–52 weeks)	Obesity Insulin resistance Steatosis NASH Fibrosis Liver tumors	Lengthy feeding time required to progress to fibrosis and liver tumors
Mouse (B6/129)	Diet-induced animal model of NAFLD (DIAMOND) [24]	High-fat/high-carbohydrate diet (42% fat, 0.1% cholesterol, high fructose/glucose in drinking water; 8–52 weeks)	Obesity Insulin resistance Steatosis NASH Fibrosis Cirrhosis Liver tumors	Lengthy feeding time required to progress to cirrhosis and liver tumors
Mouse (C57BL/6)	Fast-food diet [25]	High-fat/high-carbohydrate diet (40% fat, 2% cholesterol, high fructose in drinking water; 25 weeks)	Obesity Insulin resistance Steatosis NASH Fibrosis	Liver tumors not reported in this model
Mouse (C57BL/6)	STAM™ (streptozotocin + high-fat diet) [26]	Perinatal injection of streptozotocin followed by high-fat diet (6–20 weeks)	Obesity Insulin resistance Steatosis NASH Fibrosis Liver tumors	Only male mice developed liver tumors
Monkey (Cynomolgus)	NASH diet [27]	High-fat/high-carbohydrate/high-cholesterol diet (47% fat, 37% carbohydrates, 1% cholesterol; 10 months)	Obesity Hyperglycemia Hyperlipidemia Steatosis NASH Fibrosis	Studies were only performed in male monkeys
**Models of oxidative stress**				
Mouse	PPARγ deletion in hepatic macrophages + carbon tetrachloride [28]	Hepatic macrophage-specific deletion of peroxisome proliferator-activated receptor γ, liver fibrosis induced by carbon tetrachloride	Oxidative stress Hepatic inflammation Fibrosis	No evidence of MetS
Zebrafish	*fabp10-CETI-PIC3* + high-fat diet [29]	Over-expression of pro-inflammatory cytokines	Insulin resistance Oxidative stress Steatosis	No evidence of fibrosis or liver tumors
**Models of endoplasmic reticulum (ER) stress**				
Mouse (C57BL/6)	MUP-uPA transgenic + high-fat diet [30]	Transgenic mice expressing urokinase-type plasminogen activator (uPA) under a hepatocyte-specific promoter for major urinary protein (MUP) fed a high-fat diet (16–40 weeks)	Obesity ER stress Steatosis NASH Fibrosis Liver tumors	Lengthy feeding time required for disease progression
Zebrafish	Foie gras (fgr) [31,32]	Mutation in gene encoding an analog of trafficking protein particle complex 11 (TRAPPC11)	ER stress Steatosis NASH	Recessive lethal, does not show full pathogenesis of fibrosis or liver tumors
Zebrafish	*hi559* [33]	Mutation in *cdipt* gene encoding phosphatidylinositol synthase	ER stress Steatosis	No evidence of NASH, fibrosis, or liver tumors

## Data Availability

Not applicable.

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
