# Peer review of "Non-Alcoholic Fatty Liver Disease: Translating Disease Mechanisms into Therapeutics Using Animal Models"

_ijms, 2023, doi:10.3390/ijms24129996_

Round 1
Reviewer 1 Report
review article "Non-Alcoholic Fatty Liver Disease: Translating Disease Mechanisms into Therapeutics Using Animal Models" is rather important in the sense that NAFLD is really big problem affecting people of all ages and ethnicities, still little is known to cure or treat it. Animal models can serve important gap in research in this area to understand the diseases and treatment discovery. I suggest authors to include in detail, viral hepatitis topic also in this paper to make it more useful and comprehensive. Because this is now important, as around 4 million people around the world suffer from this condition and I believe review articles are important for giving new directions for research. Also, authors are encouraged to include future directions in this research area.
minor changes are required
Reviewer 2 Report
Excellent review, gives all the tools for people in this arena to select the right model for their needs. Very well written.
Reviewer 3 Report
Non-Alcoholic fatty liver disease (NAFLD) is characterized by an excess fat accumulation
in the liver. No current pharmacological treatment exist to manage NAFLD. This review
discuss different cellular and molecular mechanisms to contribute to NAFLD to develop
therapies.
It needs some major revisions:
1.-Line 32: fibrosis and non alcoholic steatohepatitis (NASH) are not the same. Please,
explain both disease to clarify the difference between them.
2.-In Table 1 appear steatosis but in the introduction is not well explain it, so I
recommend explaining: hepatic steatosis, fibrosis and NASH, in the section of
introduction.
3.-Line 37: Metabolic syndrome (MetS), please, add the abbreviation.
4.-Line 44: Please add the meaning of FDA
5.-Line 123: Add a reference to known where you found the definition of SREBPs.
6.-In section 5.1., lifestyle modifications, add as a lifestyle the adherence to the
Mediterranean diet. For example, the information of this article: doi:
10.3390/antiox11081440.

Round 2
Reviewer 1 Report
revised manuscript is improved
Reviewer 3 Report
Accept in present form